# The Soil Water Evaporation Process from Mountains Based on the Stable Isotope Composition in a Headwater Basin and Northwest China

**Leilei Yong [1], Guofeng Zhu [1,2,\*], Qiaozhuo Wan [1], Yuanxiao Xu [1], Zhuanxia Zhang [1], Zhigang Sun [1], Huiying Ma [1], Liyuan Sang [1], Yuwei Liu [1], Huiwen Guo [1] and Yu Zhang [1]**

[1] College of Geography and Environment Science, Northwest Normal University, Lanzhou 730070, China; nwnuyll@163.com (L.Y.); wqz2860@163.com (Q.W.); Xyxange@163.com (Y.X.); zzx_nwnu@163.com (Z.Z.); zachsuen@163.com (Z.S.); nwnumhy@163.com (H.M.); NWNUSLY@163.com (L.S.); liuyuweinwnu@163.com (Y.L.); guohuiwenl@163.com (H.G.); zhangyu1213@163.com (Y.Z.)

[2] Gansu Engineering Research Center of Land Utilization and Comprehension Consolidation, Lanzhou 730070, China

\* Correspondence: zhugf@nwnu.edu.cn

**Abstract:** Soil water is a link between different water bodies. The study of soil water evaporation is of great significance to understand the regional hydrological process, promote environmental remediation in arid areas, and rationalize ecological water use. On the basis of soil water $\delta^2$H and $\delta^{18}$O data from April to October 2017 in the Xiying River basin in the upper reaches of the Qilian mountains, the lc-excess and Craig-Gordon model were applied to reflect the evaporating fractionation of soil water. The results show that the change in evaporation loss drives the enrichment of soil water isotopes. The signal of evaporative fractionation of soil water isotopes at different elevations has spatiotemporal heterogeneity. From the perspective of time dynamics, the evaporation loss of the whole region during the observation period was affected by temperature before July, while after July, it was controlled jointly by temperature and humidity, evaporation was weakened. Soil salt content and vegetation played an important role in evaporation loss. In terms of spatial dynamics, the soil moisture evaporation at the Xiying (2097 m) and Huajian (2390 m) stations in the foothills area is larger than that at the Nichan station (2721 m) on the hillside and Lenglong station (3637 m) on the mountain top. The surface soil water evaporation is strong, and the evaporation becomes weak with the increase of depth. The research has guiding significance for the restoration and protection of vegetation in arid areas and the formulation of reasonable animal husbandry policies.

**Keywords:** Qilian Mountains; stable isotope; evaporation loss; lc-excess; Craig-Gordon model

## 1. Introduction

Soil water is the important link for hydrothermal exchange across different areas of the globe [1–3]. Soil water stable isotope composition carries important information on prevailing soil hydrological conditions and for constraining ecosystem water budgets [4–6]. Furthermore, soil water plays a key role in maintaining material transport and energy balance in a region [7–9]. The accurate calculation of soil evaporation and knowledge of its changing characteristics are of great significance for revealing soil–vegetation–atmosphere interactions [2,10–12]. A stable isotope is a natural tracer widely distributed in natural waters. Compared to traditional research methods, soil water isotope techniques can reveal particular soil hydrological process information, such as infiltration [13,14], evaporation [15–17], transpiration [18], and mixing process [19,20] more accurately and quickly. They are widely used to

analyze the dynamic changes of water sources, migration and circulation, and quantitative estimation of hydrological processes [21–24].

Water evaporation is an important part of the regional water balance. Scholars have carried out a large number of studies on water evaporation and fractionation. Craig and Gordon applied Fick's law of diffusion to simulate the isotope motion of the gas and liquid phases on both sides of the interface between the atmosphere and water and established an analytical model for the migration of isotope components from the surface of free water [25]. Majoube established the relationship between $\delta^2$H and $\delta^{18}$O fractionation coefficients and temperature under liquid–gas phase equilibrium conditions, which provided a quantitative basis for simulating the equilibrium evaporation isotope fractionation of water [26]. Whether it is Rayleigh equilibrium fractionation or dynamic fractionation, the Craig-Gordon model is regarded as the basis for calculating the stable isotope fractionation in waters [27]. Based on the improved Craig-Gordon model, Krzypek developed a software (Hydrocalculator) that quickly calculates the stable isotope value of evaporated water vapor, which greatly facilitates the calculation of hydrogen and oxygen isotopes of water evaporation [28].

In arid and semi-arid areas, evaporation is the main method of soil water consumption and one of the main driving forces of vertical soil movement [29,30]. In the 1960s, Zimmermann first used the stable isotope of hydrogen and oxygen to explore the process of soil water evaporation fractionation [31]. This groundbreaking development laid the foundation for the study of soil water evaporation using hydrogen and oxygen isotopes [32,33]. Subsequently, using isotope distribution to study soil evaporation laws developed rapidly, and extended to saturated and unsaturated soils [34,35], stable and unstable states of soils, and constant and non-constant temperature states of soils [36,37]. In many studies, the slope of the relationship between $^2$H and $^{18}$O has been often used to identify water evaporation. Using these Craig and Gordon models, the predicted slope of evaporation is between four and six under most climatic conditions, and air temperature, relative humidity, isotopic composition, and kinetic fractionation effects are the main causes of this variation [38]. Since surface runoff, soil water mixing, and plant root water uptake do not cause hydrogen and oxygen stable isotope fractionation [39,40], the fractionation mechanism of hydrogen and oxygen stable isotopes has been widely used to calculate soil surface evaporation.

Soil evaporation is an important process of water dissipation and is the key link of the entire hydrological process to inland river basins [18,29,41–45]. Disregarding the soil water evaporation process will inevitably lead to the cognitive deviation of the water cycle process which will ultimately limit the study of the regional hydrological process [12,13]. Taking the Xiying River Basin as an example, we used the isotope evaporation model to study the law of soil evaporation in high mountain areas. We hope this will improve our understanding of the water cycle process, and be beneficial for estimating soil evaporation on different underlying surfaces in mountainous areas. What's more, this research will provide a scientific and reasonable theoretical basis for the use of water resources in the fragile ecological environment and the formulation of animal husbandry policies.

## 2. Data and Methods

### 2.1. Study Area

The Qilian Mountains are the source of more than 100 rivers, one of which, the Xiying River has a large flow and is located in the eastern part of the mountain (101°40′47″~102°23′5″ E, 37°28′22″~38°1′42″ N). The Shuiguan, Ningchang, Xiangshui, and Tuta rivers flow into the Xiying river from the southwest to the northeast, and finally, it flows into the Xiying reservoir. In spring, as temperatures rise and snow melts, the flood season begins, reaching a peak in summer, when precipitation is at its highest. The study area is located in the alpine and semi-arid areas of the Qilian Mountains, at an altitude of 2000 to 5000 m. This basin has a continental temperate arid climate with strong solar radiation, long sunshine hours, and large diurnal temperature differences. Annual precipitation ranges from 300 to 600 mm, but annual evaporation is also high, varying from

700 to 1200 mm. In the upper reaches of the basin, there are clear vegetation zones of mainly temperate coniferous forest, alpine azalea shrub, and desert steppe vegetation. The middle and lower reaches are mainly bare land, with little vegetation coverage. Soils are mainly lime, chestnut, alpine scrub meadow, and desert soil (Figure 1).

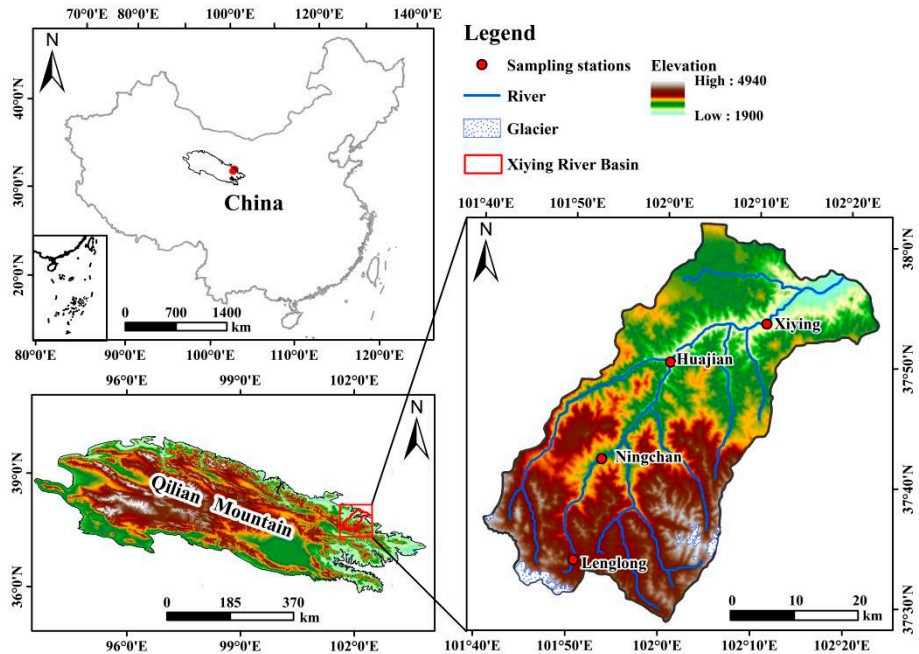

**Figure 1.** Overview of the study area and location of sampling points.

## 2.2. Sample Collection

From April to October 2017, samples of precipitation and soil were collected at Xiying, Huajian, Ningchan, and Lenglong (Table 1).

Precipitation sample collection: precipitation samples were collected using a plastic funnel bottle device. To prevent evaporation, a table tennis ball was placed above the funnel mouth and the precipitation was collected immediately after each precipitation event. The sample was transferred to a 100 mL high-density polyethylene sample bottle, sealed with a Parafilm membrane, and then frozen and stored until experimental analysis.

Soil sample collection: soil samples were drilled with soil drills, and samples were taken every 10 cm to collect fresh soil samples in the 100 cm soil layer. The collected soil sample was placed into a 100 mL glass bottle, the bottle mouth was sealed with Parafilm marking the sampling dates, then frozen and stored until experimental analysis.

**Table 1.** Basic data of each sampling station (Long—Longitude, Lat—Latitude, Alt—Altitude, T—Air Temperature, P—Precipitation Amount, h—Relative Humidity).

| Sampling Station and Abbreviation | | Geographical Parameter | | | Meteorological Parameters | | | Number of Samples | | Vegetation Species |
|---|---|---|---|---|---|---|---|---|---|---|
| | | *Long* (°E) | *Lat* (°N) | *Alt* (m) | *T* (°C) | *P* (mm) | *H* (%) | Precipitation | Soil | |
| M1 | Lenglong | 101°51′16″ | 37°33′28″ | 3637 | −0.19 | 595.1 | 69.2 | 72 | 47 | *Alpine meadow* |
| M2 | Ningchan | 101°53′23″ | 37°41′50″ | 2721 | 3.34 | 431.9 | 66.6 | 42 | 41 | *Picea crassifolia* |
| M3 | Huajian | 102°00′25″ | 37°50′23″ | 2390 | 6.6 | 363.5 | 60.4 | 37 | 54 | *Subalpine shrub* |
| M4 | Xiying | 102°10′56″ | 37°53′27″ | 2097 | 7.9 | 262.5 | 59.8 | 40 | 53 | *Populus* |

## 2.3. Sample Processing and Analysis

The processing and determination of the samples were all completed in the isotope laboratory of the College of Geography and Environmental Science, Northwest Normal University. Water from the

soil was extracted using an LI-2100 fully automated cryogenic vacuum extraction system, all water samples were analyzed using a DLT-100 liquid water isotope analyzer developed by Los Gatos Research, USA. All samples and isotopic standards were injected sequentially six times using a microliter syringe, and the average of the last four injection results was accepted as the final value with the first two injections being discarded. The analytically derived $\delta^2$H and $\delta^{18}$O are expressed in terms of the thousandth difference from the Vienna Standard Mean Ocean Water:

$$\delta = \left( \frac{R_{\text{sample}}}{R_{\text{standard}}} - 1 \right) \times 1000‰ \tag{1}$$

In the formula, the $R_{\text{sample}}$ is the ratio of heavy isotope to light isotope in the water sample. $\delta^2$H and $\delta^{18}$O are $^2$H/H and $^{18}$O/$^{16}$O. The standard error of measurement is ±0.6‰ for $\delta^2$H and ±0.2‰ for $\delta^{18}$O, respectively.

*2.4. Meteorological Data*

Meteorological data were passed through the China National Meteorological Data Network (http://data.cma.cn/) and the four meteorological stations set up by Northwest Normal University in Xiying, Huajian, Ningchan, and Lenglong. The main data obtained are temperature, precipitation, and humidity.

*2.5. lc-Excess*

The linear relationship between $\delta^2$H and $\delta^{18}$O in precipitation and soil water is defined as the LMWL (local meteoric water line) and SWL (soil water line), respectively, which is of great significance for studying the evaporative fractionation of stable isotopes during the water cycle. For each soil water and precipitation sample, we further calculated the line-conditioned excess. The lc-excess in different water bodies can characterize the evaporation index of different water bodies relative to local precipitation [46]:

$$\text{lc} - \text{excess} = \delta^2\text{H} - a \times \delta^2\text{H} - b \tag{2}$$

In the formula, *a* and *b* are the slope and intercept of LMWL, respectively, $\delta^2$H and $\delta^{18}$O are the hydrogen and oxygen isotope values in the sample. Based on the precision of the isotope analysis and the slope of the LMWL, the lc-excess of soil water and rainfall was derived, and their uncertainty was estimated to be ±2.27‰. The physical meaning of lc-excess is expressed as the degree of deviation of the isotope value in the sample from the LMWL, indicating the non-equilibrium dynamic fractionation process caused by evaporation [47]. Generally, the change of lc-excess in local precipitation is mainly affected by different water vapor sources, and the annual average is 0. Since the stable isotopes in soil water are enriched by evaporation, the average lc-excess is usually negative [48].

*2.6. Craig-Gordon Model*

We estimated the evaporative losses *f* based on the Craig-Gordon model and isotope mass balance equation [27,32], the formula is as follows:

$$f = 1 - \left[ \frac{(\delta_s - \delta^*)}{(\delta_p - \delta^*)} \right]^m \tag{3}$$

In the formula, $\delta_s$ is the soil water isotope value of each soil layer. $\delta_p$ is the isotopic composition of the original water source, calculated by the intersection of LMWL and SEL (Soil water Evaporation Line) in the open liquid-vapor isotope system [49], the formula is as follows:

$$\delta^{18}\text{O}_{\text{intersect}} = \frac{n - b}{a - m} \tag{4}$$

$$\delta^2 H_{intersect} = a\delta^{18}O_{intersect} + b \tag{5}$$

Among them, $a$ and $b$, and m and n are the slope and intercept of LMWL and SEL, respectively. $\delta^*$ is the limiting factor for isotope enrichment [27], which can be calculated by the following formula:

$$\delta^* = \frac{h\delta_A - \varepsilon_k + \varepsilon^+/\alpha^+}{h - 10^{-3}(\varepsilon_k + \varepsilon^+/\alpha^+)} \tag{6}$$

In the formula, $h$ is the relative humidity in the air. $\delta_A$ is the surface water vapor isotope value, the calculation method is:

$$\delta_A = \frac{\delta_{rain} - k\varepsilon^+}{(1 + k\alpha^+ \times 10^{-3})} \tag{7}$$

In the formula, $\delta_{rain}$ is corrected by local evaporation line (LEL) [28], $k$ is an adjustment parameter (it is the difference between the measured value of the LEL), $\varepsilon^+$ is the isotope fractionation factor between water and vapor. The formula is as follows:

$$\varepsilon^+ = \left(\alpha^+ - 1\right) \times 1000 \tag{8}$$

$\alpha^+$ is the equilibrium fractionation factor depending on temperature, Horita and Wesolowski [50] give a calculation for $\alpha^+$. Equation (9) calculates $\alpha^+$ of $^2H$, and Equation (10) calculates $\alpha^+$ of $^{18}O$:

$$10^3 \ln^2 \alpha^+ = \frac{1158.8T^3}{10^9} - \frac{1620.1T^2}{10^6} + \frac{794.84T}{10^3} - 161.04 + \frac{2.9992 \times 10^9}{T^3} \tag{9}$$

$$10^3 \ln^2 \alpha^+ = -7.685 + \frac{6.7123 \times 10^3}{T} - \frac{1.6664 \times 10^6}{T^2} + \frac{0.35041 \times 10^9}{T^3} \tag{10}$$

Among them, $T$ stands for temperature (K), then the LEL slope ($S_{LEL}$) can be calculated as:

$$S_{LEL} = \frac{\left[\frac{h(10^{-3}\delta^2 H_A - 10^{-3}\delta^2 H_{rain}) + (1 + 10^{-3}\delta^2 H_{rain})10^{-3}\varepsilon}{h - 10^{-3}\varepsilon}\right]_H}{\left[\frac{h(10^{-3}\delta^{18} H_A - 10^{-3}\delta^{18} H_{rain}) + (1 + 10^{-3}\delta^{18} H_{rain})10^{-3}\varepsilon}{h - 10^{-3}\varepsilon}\right]_O} \tag{11}$$

where $\varepsilon$ is the total fractionation factor defined as [28]:

$$\varepsilon = \varepsilon^+/\alpha^+ + \varepsilon_k \tag{12}$$

where kinetic fractionation factor ($\varepsilon_k$) [51] is calculated as:

$$\varepsilon_k = (1 - h)n\theta C_D \tag{13}$$

where $\theta$ is the ratio of the molecular diffusion fractionation coefficient to the total diffusion fractionation coefficient. For water bodies where evaporation flux does not significantly interfere with humidity, including soil evaporation, it is generally taken as 1; $n$ is constant, it is about the correlation between molecular diffusion resistance and molecular diffusion coefficient. For non-flowing gas layers (soil evaporation and plant transpiration), it is generally taken as 1 [28]; $C_D$ is a parameter describing the molecular diffusion efficiency, the values are 25.1‰ for hydrogen and 28.5‰ for oxygen, respectively [52].

The calculation process is repeated for values of $k$ between 0.6 and 1.0 with a step width of 0.0001; the value of $k$ depends on the difference between the calculated slope of LEL and the observed slope of LEL. When the difference between the two was the lowest (or $k$ reached the boundary values, 0.6 or 1.0), the final $k$ value was taken [28].

Finally, *m* is the enrichment slope as defined previously [53,54].

$$m = \frac{h - 10^{-3}(\varepsilon_k + \varepsilon^+/\alpha^+)}{1 - h + 10^{-3}\varepsilon_k} \tag{14}$$

## 3. Results

### 3.1. The Isotopic Composition of Various Water Bodies and Its Evaporation Signal

The precipitation stable isotope is the input signal of various water bodies in the global water circulation process. Changes in the isotopic composition will directly affect the distribution of isotopic concentrations in surface water, soil water, and groundwater. The distribution of precipitation isotope $\delta^2H$ in the Xiying River Basin ranged from −163.9‰ to 23.2‰, and the distribution of $\delta^{18}O$ ranged from −23.1‰ to 3.2‰. The mean value of $\delta^2H$ and $\delta^{18}O$ decreased from the foothills to the top of the mountain, which reflects the precipitation isotope that decreased with increasing altitude in the area. The distribution range of soil water ($\delta^2H$: −84.3‰−−18.5‰, $\delta^{18}O$: −13.5‰–2.9‰) was significantly smaller than precipitation, and the average value of soil water isotopes increased from the top to the foothills (Table 2). This is mainly due to the fact that as the altitude decreases, the increase in evaporation makes the isotope fractionation significant. Of course, evaporation is not the only factor that also affects the composition of soil water isotopes. From the top of the mountain to the foothills, the amount of precipitation gradually decreases, and the precipitation isotope value gradually increases. The influence of water mixing on the soil water isotope during the precipitation process at each station cannot be ignored.

**Table 2.** Isotopic composition of each water at four stations in the Xiying River basin.

| Stations | Types | $\delta^2H$/‰ | | | $\delta^{18}O$/‰ | | |
|---|---|---|---|---|---|---|---|
| | | Max | Min | Average | Max | Min | Average |
| Lenglong | Precipitation | 13.7 | −163.9 | −73.1 | −1.3 | −23.1 | −10.0 |
| | Soil | −42.4 | −83.9 | −63.5 | −6.1 | −12.6 | −9.3 |
| Ningchan | Precipitation | 13.0 | −117.8 | −42.0 | −0.1 | −17.4 | −7.1 |
| | Soil | −18.5 | −78.45 | −58.2 | −2.1 | −12.0 | −8.1 |
| Huajian | Precipitation | 4.2 | −103.1 | −37.4 | −0.9 | −15.1 | −5.9 |
| | Soil | −20.1 | −74.9 | −45.3 | 2.9 | −11.8 | −4.7 |
| Xiying | Precipitation | 23.2 | −110.2 | −31.8 | 3.2 | −15.2 | −5.8 |
| | Soil | −18.5 | −84.3 | −54.8 | 0.2 | −13.5 | −6.9 |

In order to better analyze the environmental significance of precipitation and soil water indicators and the replenishment relationship between the two, according to the $\delta^2H$ and $\delta^{18}O$ values, the least square method is used to fit the LMWL and SWL of each sampling point in the Xiying River Basin (Figure 2). It can be seen from the figure that there is a good linear relationship between $\delta^2H$ and $\delta^{18}O$ in atmospheric precipitation at each sampling point. Due to differences in natural environment factors such as topography, underlying surface, climate, etc., the local atmospheric waterline differed from place to place. At the Lenglong station, which has the highest altitude, the slope and intercept of LMWL were higher than those of GMWL (the global meteoric water line). On the one hand, the local temperature, cloud base height and air saturation water vapor loss were lower, so the secondary evaporation under the cloud was lower, on the other hand, monsoons caused strong convective precipitation. The slope of LMWL of the other three sampling points was lower than that of GMWL. Meteorological factors and topography will affect the cloud height and saturation deficit in high altitude areas, and sub-cloud evaporation will be strengthened as the altitude decreases. Therefore, the relatively high temperature and strong sub-cloud evaporation caused the slope and intercept of the LMWL to gradually decrease from the top to the foothill.

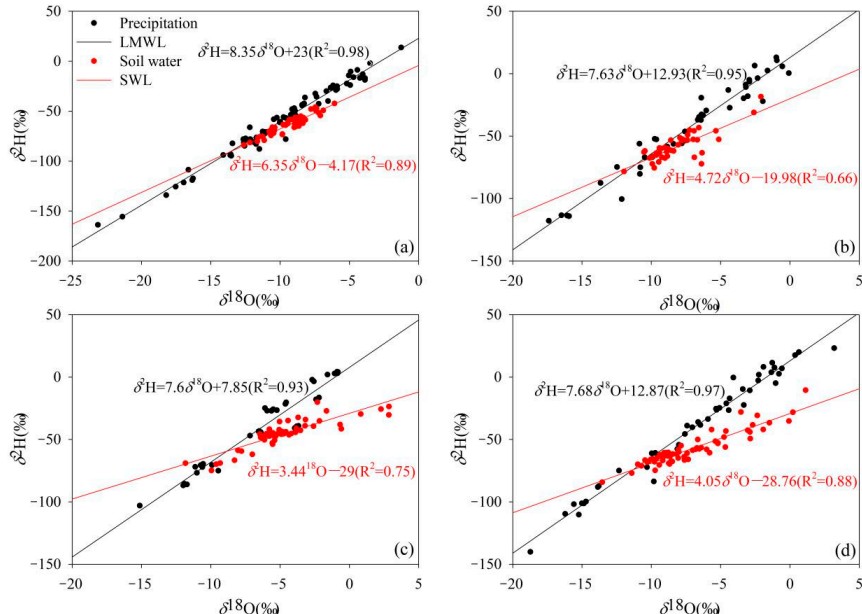

**Figure 2.** Relationship between soil water $\delta^2$H and $\delta^{18}$O at each sampling point ((**a**) Lenglong; (**b**) Ningchan; (**c**) Huajian; (**d**) Xiying).

The $\delta^2$H and $\delta^{18}$O of soil water at each sampling point were mostly located at the lower right of LMWL, which indicated that atmospheric precipitation is the main source of soil water replenishment and soil water is subject to different degrees of evaporation. From the Lenglong station at the top of the mountain to the Huajian station at the foot of the mountain, the slope and intercept of the SWL decreased. As the altitude decreases, the temperature rises, and the relative humidity of the atmosphere decreases, and the evaporation of soil moisture gradually increases. Due to the influence of vegetation, the two sampling points in the foothills broke the rule of change with altitude. Compared with the lower altitude of the Xiying station (woodland), the surface of the Huajian station was covered with sparse shrubs. A large area of the exposed ground causes soil evaporation to be more intense than the former (Figure 2).

*3.2. Temporal Dynamics in Soil Water Isotopes*

During the sampling period, the Lenglong station had the most precipitation events (72 times) and the monthly distribution was relatively uniform. The other three stations had relatively few precipitation events, concentrated in late July to mid-September. The atmospheric precipitation $\delta^{18}$O at the four stations (due to the same changes in $\delta^2$H and $\delta^{18}$O, so this article only analyzes the changes in $\delta^{18}$O) has a similar trend. From April to August, the $\delta^{18}$O value fluctuates increased, reaching the maximum around mid-August, and then gradually decreased. The high $\delta^{18}$O value mostly occured in August (Lenglong: −1.3‰ (4 August); Ningchan: −0.1‰ (August 10); Huajian: −0.8‰ (7 August); Xiying: 3.2‰ (13 August)) (Figure 3). The lc-excess of the Lenglong station and Ningchan station was mostly negative from April to August, and mostly positive after September. Low values mostly occurred in spring and summer, and during this period of precipitation, the effect of secondary evaporation under clouds was more obvious. The lc-excess of Huajian station was mostly negative before July, and the value was higher after July, and the low value mostly appeared in June (−19.0‰). The lc-excess of the Xiying station was mostly negative from April to September and only appeared positive in October. The precipitation process during the observation period was subject to different degrees of secondary evaporation (Figure 4).

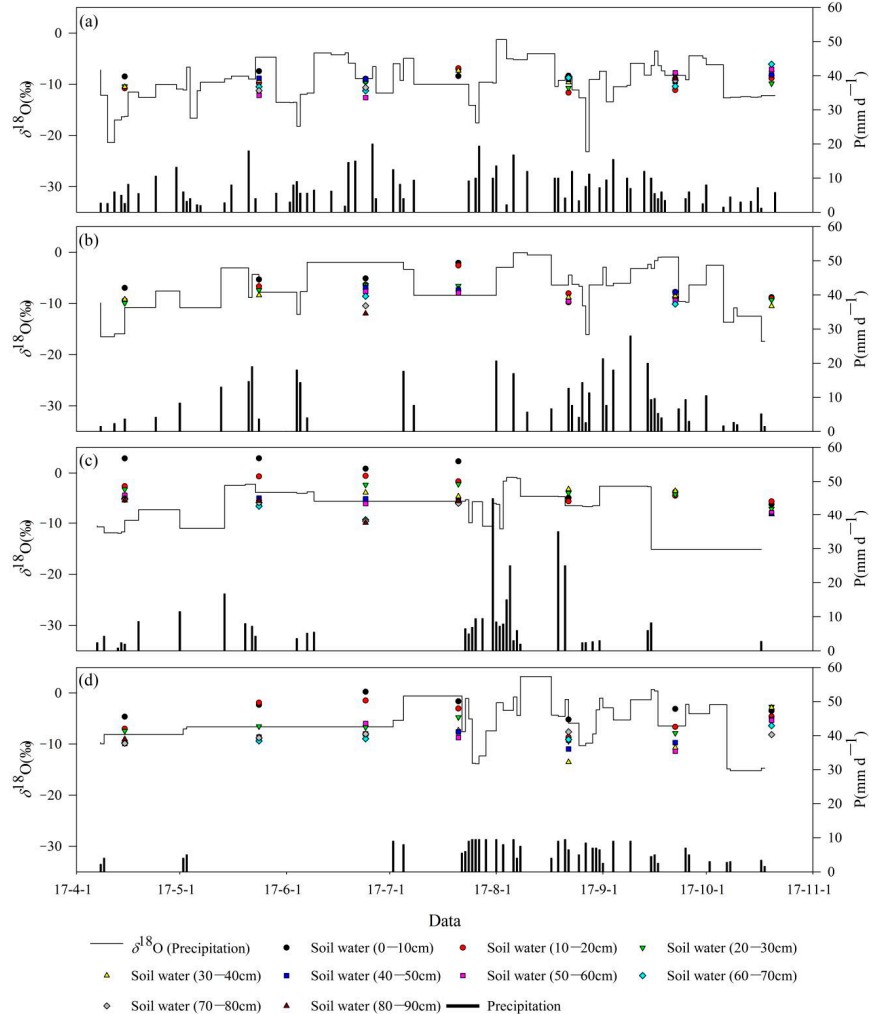

**Figure 3.** $\delta^{18}$O values for water samples of soil water at different depths and precipitation at the four stations ((**a**) Lenglong; (**b**) Ningchan; (**c**) Huajian; (**d**) Xiying).

Seasonal changes in precipitation affect soil water isotope changes at each station. The Lenglong station at the top of the mountain is rich in precipitation, cold in temperature, and the deep soil is frozen all year round. During the observation period, the mean value of $\delta^{18}$O of soil water was relatively high in July ($-7.5‰$) and October ($-7.9‰$), and the remaining months were lower. The highest value appeared in October, with a value of $-6.1‰$ (60–70 cm). There is little change from month to month. The lc-excess of each soil layer was less than zero at this station, but the change between the soil layers was relatively small. Contrary to the monthly change of soil water $\delta^{18}$O, the average value of lc-excess is in July ($-11.5‰$) and October ($-14.9‰$) was relatively low, and the evaporation and fractionation of soil water was strong in these two months. The average value of $\delta^{18}$O of soil water at the Ningchan station increased slowly from April to July, reaching the peak value ($-5.6‰$) in July. Due to the input of seasonal precipitation in August, the $\delta^{18}$O value of soil water decreased rapidly and stabilized in the following months. Affected by the melting of snow and ice and biological activities (grazing), the lc-excess of some soil layers at the site was positive in June and October. During the sampling period, the monthly average value of lc-excess first decreased and then increased, reaching the lowest value ($-13.7‰$) in June. The strength of soil evaporation changes with the change in regional temperature. The $\delta^{18}$O values of 0 to 10 cm soil water at the Huajian station from April to July were all greater than zero (April: $2.8‰$; May: $2.9‰$; June: $0.8‰$; July: $2.3‰$). Similar to the Ningchan station, the input of seasonal heavy rainfall from late July to mid-August mixed with old water caused the shallow soil water $\delta^{18}$O to be relatively depleted in August. As the soil depth

increases, this effect gradually decreases. The average value of lc-excess in each month was higher than that of other sites, but there was no obvious inter-month change. The 0 to 20 cm soil layer has obvious changes. The lc-excess value is less than −20‰ from April to July, especially the 0 to 10 cm soil layer, the lc-excess value is close to −40‰, which reflects that the shallow soil water evaporation fractionation is strong during this period. After July, the lc-excess value is not less than −20‰, and the soil water evaporation is weakened. The soil water $\delta^{18}O$ average value of the Xiying station gradually increased to the peak value (−5.5‰) from April to July, and rapidly decreased to the valley value (−9.2‰) in August, and gradually increased to October after August. The change of soil water lc-excess was the opposite. The minimum value appeared in July (−24.5‰), and the other low value occurred in October (−23.9‰). It can be seen from the change of lc-excess that there was less evaporation in August and September. In general, the $\delta^{18}O$ value of soil water fluctuated and increased from April to July, and the high value mostly appeared in July. After July, the $\delta^{18}O$ value became depleted. This rule was an exception at the Lenglong and Xiying stations. Both stations showed high $\delta^{18}O$ in October (Figure 3). The change in lc-excess is the opposite. The lc-excess decreased from April to July and the low value mostly occurred in July. Both Lenglong and Xiying stations had low lc-excess in October (Figure 4).

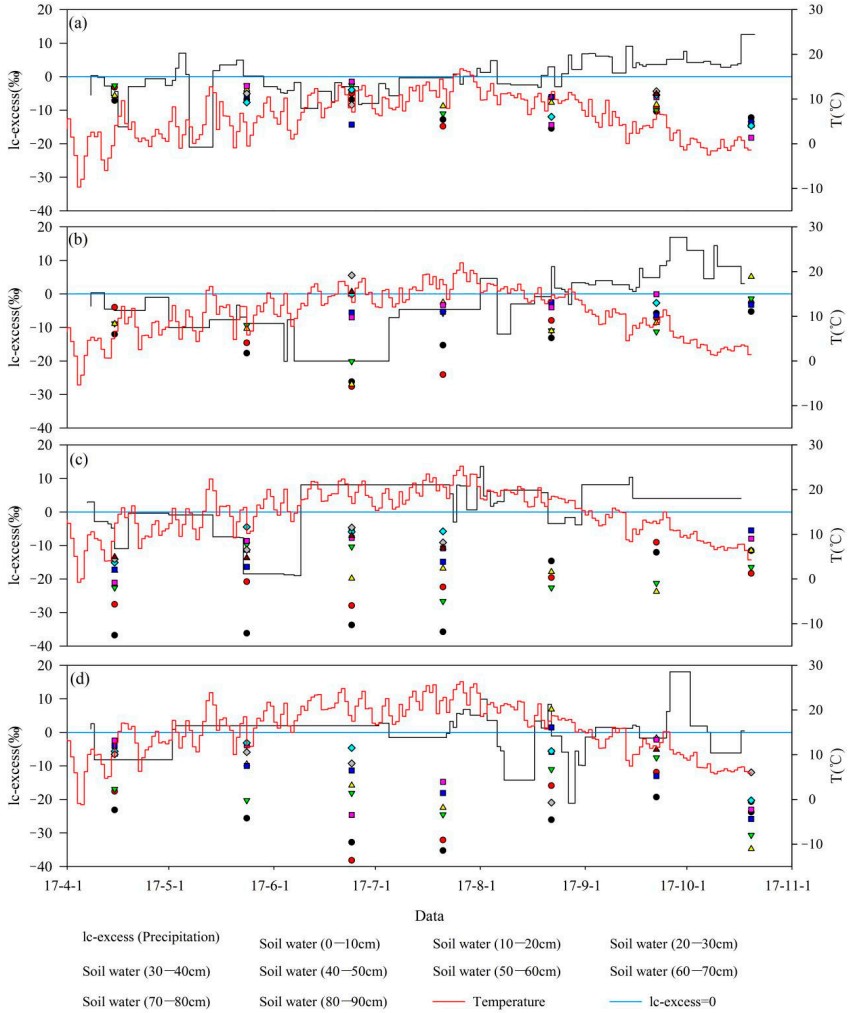

**Figure 4.** Temperature, line-conditioned excess (lc-excess, in ‰) values for water samples of soil water at different depths and precipitation at the four stations in the study area ((**a**) Lenglong; (**b**) Ningchan; (**c**) Huajian; (**d**) Xiying).

### 3.3. Spatial Soil Water Isotope Patterns

The comparison of $\delta^{18}O$ and lc-excess of soil water within 0 to100 cm of different sampling points showed that there are significant differences between soil water at different sampling points (Figures 5 and 6). During the entire sampling period, the difference in soil isotope values between different locations was most obvious at 0 to 10 cm on the surface. Compared with the Huajian (2390 m) and Xiying (2097 m) stations at the foot of the mountain, the $\delta^{18}O$ values of the Lenglong station (3637 m) on the top of the mountain and Ningchan station (2721 m) on the mountainside were obviously smaller, and the lc-excess was larger. The monthly average was $\delta^{18}O < -8.4‰$ for each soil layer at the Lenglong station, and the lc-excess $> -10.3‰$. The main source of soil water is atmospheric precipitation. The altitude effect of precipitation isotope makes the isotope value smaller in high altitude areas, which leads to the overall lower soil water value at this station. The geography and natural environment of a low temperature make each soil layer less subject to dynamic fractionation. In addition, the soil layers are seasonally freezing, and weak evaporation most affects the surface soil. Interestingly, the $\delta^{18}O$ in the profile of soil water at the Lenglong station had a unique change in early autumn (September). The strong fluctuation of the surface soil deepened to 60 cm and the highest value of the entire soil layer also appeared at 60 cm. The reason for this phenomenon is that after a relatively high-temperature summer, the seasonally frozen soil layer is in a melting state, and the evaporation signal can penetrate deep into the middle and lower soil. In addition, the high-frequency precipitation before sampling makes precipitation mixed with meltwater also caused this phenomenon. On most sampling dates at the Ningchan station, as the soil depth increased, the $\delta^{18}O$ value became more depleted. The monthly average $\delta^{18}O$ value of the 0 to 10 cm soil level was $-6.6‰$. As the soil layer deepens, it is only $-12.0‰$ at 80 to 90 cm. On the contrary, as the soil depth increased, the lc-excess value gradually increased, from $-13.6‰$ of 0 to 10 cm to $5.6‰$ of 80 to 90 cm. This change is mainly due to kinetic fractionation. During the evaporation process, the light isotopes evaporate first, so the remaining water is rich in heavy isotopes. The surface soil is susceptible to the strong influence of the climate. As the depth increases, the dynamic fractionation gradually decreases, and the isotopic composition of the soil water is gradually depleted. The Huajian and Xiying stations are located in the foothills. Compared with higher altitude areas, the $\delta^{18}O$ value was obviously higher, and the lc-excess value was lower. The monthly average values of soil water $\delta^{18}O$ in the 0 to 10 cm soil layer of the two stations were $-0.8‰$ (Huajian) and $-2.9‰$ (Xiying), and the monthly average values of lc-excess were $-25.8‰$ (Huajian) and $-26.6‰$ (Xiying), respectively. Similar to other points, shallow soil and middle soil had obvious changes, and deep soil changes tended to be stable. This is due to the strong evaporation of the two places and the strong dynamic fractionation of the surface layer. As the depth of the soil layer increases, the dynamic fractionation weakens. The $\delta^{18}O$ and lc-excess of the deep soil layer tend to be constant.

### 3.4. Evaporation Estimates

Our research results clearly show the composition and change of soil water isotope under the geographical environment of different altitudes in high mountains. On this basis, the mathematical model was used to estimate the evaporation losses of the 0 to 40 cm soil layer at four stations in the Xiying River Basin during the observation period (Figure 7). From the perspective of time changes, the average evaporation loss $f$ fluctuated and increased from April to July, and it reached an extreme value in July (Lenglong, 37.0%; Ningchan, 41.5%; Huajian, 62.4%; Xiyign, 50.0%). It decreased rapidly after July, and the evaporation became weaker. This change was not absolute. The Lenglong and Xiying stations in October were exceptions. The evaporation losses of these two stations increased in October (Lenglong, 24.1%; Xiying, 56.3%). At this time, the average evaporation loss $f$ at the Lenglong station was only less than July, and the average evaporation loss at the Xiying station was the largest in the entire observation (Figure 8). From the perspective of spatial changes, the evaporation losses of the Lenglong and Ningchan stations were significantly smaller than those of the Huajian and Xiying stations. Of course, in different months, each station also had a different intensity of evaporation.

For example, in October, the *f* of the Lenglong station was greater than that of the Ningchan and Huajian stations. For different soil layers of 0 to 40 cm, the soil water at 0 to 10 cm was generally higher than other soil layers. The evaporation loss of soil moisture below 10 cm rarely exceeded 0 to 10 cm during the sampling process.

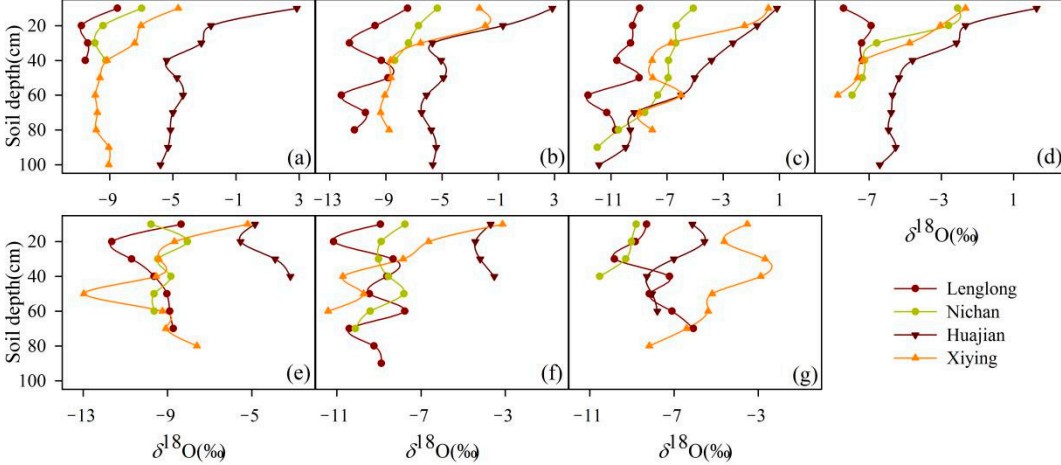

**Figure 5.** Profile variation of soil water $\delta^{18}O$ at the four stations in the study area ((**a**) Apr; (**b**) May; (**c**) Jun; (**d**) July; (**e**) Aug; (**f**) Sept; (**g**) Oct).

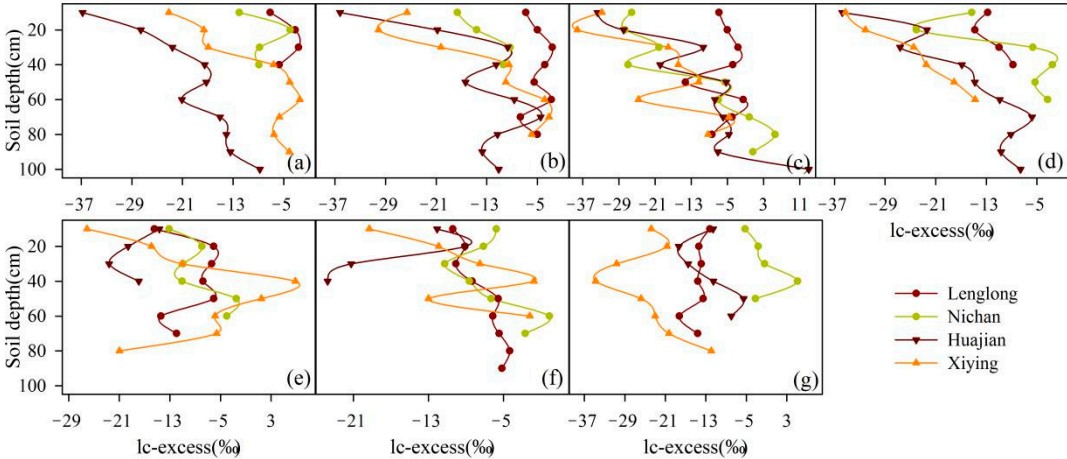

**Figure 6.** Profile variation of soil water lc-excess at the four stations in the study area ((**a**) Apr; (**b**) May; (**c**) Jun; (**d**) July; (**e**) Aug; (**f**) Sept; (**g**) Oct).

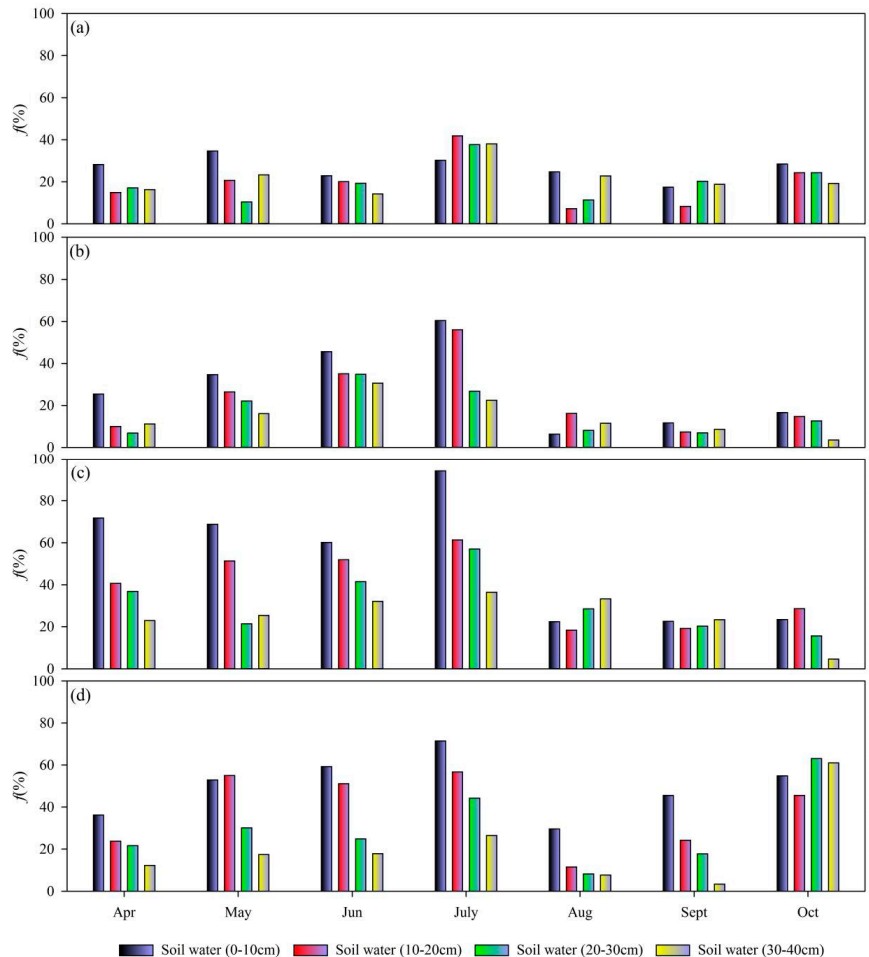

**Figure 7.** Monthly change of the 0 to 40 cm soil layer evaporation loss *f* at the four stations from April to October based on $\delta^{18}$O ((**a**) Lenglong; (**b**) Ningchan; (**c**) Huajian; (**d**) Xiying).

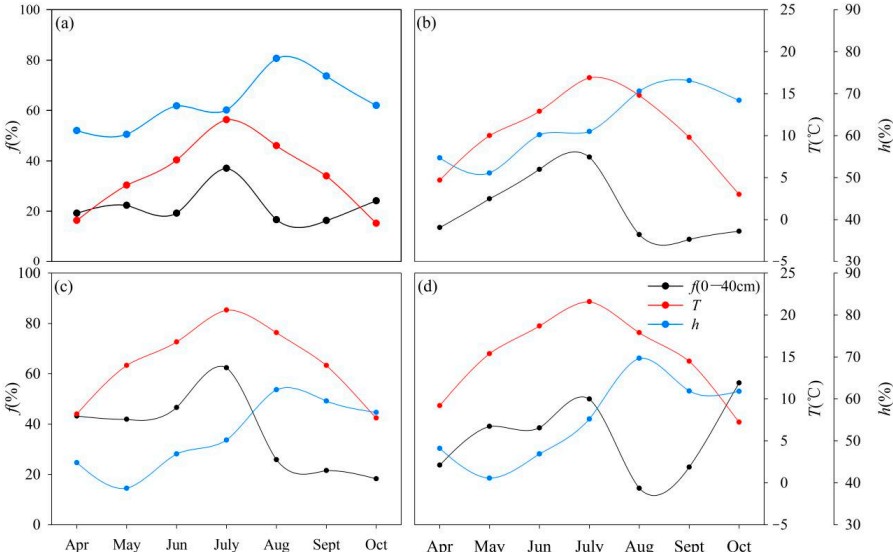

**Figure 8.** Changes in temperature, relative humidity, and 0 to 40 cm mean value of evaporation loss *f* at the four stations during the observation period ((**a**) Lenglong; (**b**) Ningchan; (**c**) Huajian; (**d**) Xiying).

## 4. Discussion

### 4.1. Temperature and Humidity Affects Soil Water Evaporation Losses

Evaporation is the process in which the kinetic energy of the thermal motion of the molecules in a liquid reaches a certain level, breaking free from the liquid surface and entering the air [51,55–58]. In the study of water evaporation, the temperature is always the main factor affecting water evaporation [29]. The diffusion coefficient of water vapor, saturated water vapor density, and hydraulic conductivity all depend on temperature and its change, accordingly [59]. Theoretically, there is a positive correlation between water evaporation intensity and air temperature [21,29]. As far as the entire watershed is concerned, the temperature $T$ and the evaporation loss $f$ of soil moisture show a positive correlation (Figure 9), which is similar to the theoretical and past research results [13,36,60]. The four soil sampling sites have a large altitude difference. The lower the altitude the higher the temperature, and the temperature will accelerate the irregular movement rate of water molecules, which will lead to stronger evaporation [34]. During the observation period, the temperature at each station began to rise from April. Among them, the temperature at the Xiying (9.2 °C) and Huajian (8.2 °C) stations was higher, followed by the Ningchan station (4.7 °C), and the Lenglong station had the lowest temperature of only −0.1 °C. In July, the temperature at each station rose to the maximum, from the top to the foot of the mountain, they were 11.9, 16.9, 20.6, and 21.6 °C, respectively, and began to decrease in August. By October, the temperature at the Huajian station (7.7 °C) was the highest. The average loss of soil moisture from 0 to 40 cm at the Lenglong, Ningchan, and Huajian stations changes similar to temperature. The evaporation loss increases from April to July, the evaporation loss $f$ in July is significantly higher than that in the other months and starts to decrease in August. The Xiying station has different trends. The station has two rising periods during the observation period. The period from April to July is consistent with other stations, which is a period of increased evaporation loss, decreased in August, and another period of rising in September and October. Che found similar seasonal changes in the western Loess Plateau [61]. Comparing different stations, the evaporation loss $f$ at the Huajian and Xiying stations is relatively large, while the Lenglong and Ningchan stations are relatively small. From this point of view, the change trend of the evaporation loss $f$ at different sites in the Xiying River Basin during most months of the observation period is essentially the same as the temperature, but the degree of influence of the temperature on evaporation at different sites is different. For example, during August and October, the temperature at the Huajian station was not much different from that at the Xiying station, but the evaporation loss was significantly different between the two stations. Similarly, although the temperature at the Lenglong station is lower than that at the Ningchan station, the evaporation loss of the former is higher than that of the latter after August (Figure 8) [26].

The higher the relative humidity of the atmosphere, and the closer the water vapor pressure is to the saturated state, the more water vapor will be captured, and the less obvious the phenomenon of evaporation [51]. Therefore, the relative humidity of the atmosphere is negatively correlated with evaporation. The results of our research are similar to this (Figure 9) [32]. During the observation period, the relative humidity of each station showed a trend of almost synchronous change, with a slight decrease from April to May. After May, the humidity gradually rose to its peak in August, and then began to decrease after August. The relative humidity in the Xiying River Basin is relatively small from April to June (defined as the dry season of the observation period), and relative humidity is relatively high from July to October (defined as the wet season of the observation period). The change trend of humidity and evaporation loss in the dry season is similar, which is contrary to theory, indicating that the influence of humidity on evaporation during this period is weak or is covered by other factors [42]. In the wet season, the trends of the two are opposite, and the inhibitory effect of humidity on evaporation is gradually manifested (Figure 8).

In summary, the evaporation loss before July is mainly affected by temperature, and after July, the temperature and humidity are jointly controlled. We also found that although there is a correlation

between temperature, humidity, and evaporation intensity, this correlation is weak. On the one hand, the degree of influence of temperature and humidity on the evaporation intensity varies in different periods and at different sites. On the other hand, in some sampling periods, temperature and humidity are not the main factors determining the evaporation intensity. Other factors (e.g., vegetation and human activities) directly or indirectly affect evaporation by changing the surface environment. For example, grazing causes the reduction of surface plants, and the bare ground increases evaporation. At this time, temperature or humidity is no longer the main factor affecting evaporation, which can reasonably explain the discrete points in Figure 9.

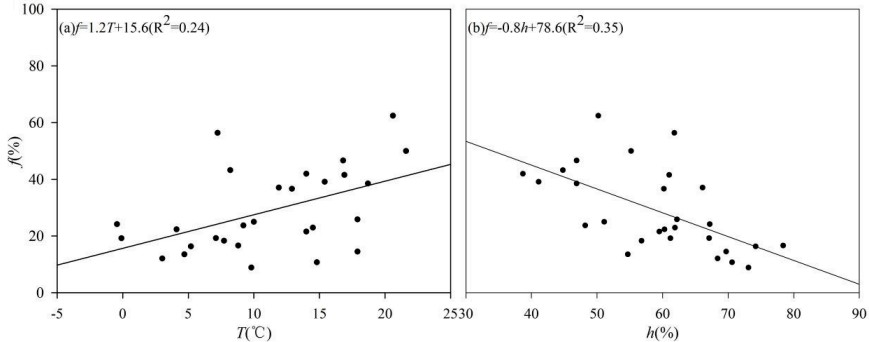

**Figure 9.** Correlation between temperature, humidity, and evaporation loss ((**a**) The relationship between temperature and evaporation loss; (**b**) The relationship between humidity and evaporation loss).

### 4.2. Soil Salt Content and Surface Vegetation Affects Soil Water Evaporation Losses

In addition to temperature and relative humidity, soil water evaporation is also restricted by soil addsalt content and surface vegetation [29,59]. Salt effects evaporation by reducing water chemical activity, thereby reducing vapor pressure, and lowering the evaporation rate. Besides, compared to pure water without salt, the higher the salt content is, the higher the surface water evaporation temperature becomes [13,29]. Studies have shown that dissolved salts have a certain inhibitory effect on equilibrium fractionation [62]. Although the amount of salt cannot directly affect equilibrium fractionation, increasing salinity can indirectly reduce it because of the proportional relation between the dynamic fractionation coefficient and relative air humidity [13,34]. Furthermore, the effect of particular kinds of salt on the equilibrium fractionation may be different [42]. Additionally, when soil water content is between the field water holding capacity and capillary water content, soil salinity has a great impact on soil water evaporation. When soil water content is below the capillary water content, soil water evaporation is mainly controlled by soil water content [59,62].

The presence or absence of surface vegetation and the different types of vegetation also affect the shape of the soil water isotope profile and further affect soil water evaporation [63]. Studies have shown that a bare surface is more beneficial for isotope enrichment than the vegetation-covered areas and that stable isotopes migrate through the water to the upper soil, resulting in enrichment [18]. This is mainly because a bare surface is more susceptible to evaporation. There are significant differences in evapotranspiration rates between different types of vegetation and for plants of different rooting depths [49]. Soil isotope profiles are different due to the different soil layers that roots can reach [36]. Vegetation with different coverage grows on the surface of the four sites at different altitudes. There are sparse alpine meadows at the surface of the Lenglong station. Although the station is located on the top of the highest altitude, compared with the Ningchan station with Picea crassifolia growing on the hillside, the evaporation loss of the former is greater than that of the latter after July. The more significant effects of plants on soil water evaporation occurred at the Huajian and Xiying stations. As we all know, woody plants have stronger water protection than herbaceous plants [10,16]. The surface soil of the Xiying station is covered by dense branches and leaves of poplar, which can effectively reduce water evaporation. However, in the Huajian station, due to the sparse subalpine shrub growing on the surface, the soil is exposed in large areas, so the water evaporation is stronger. Similarly, the effect of

vegetation on evaporation can reasonably explain the strong evaporation loss of the Xiying station in October. As a deciduous broad-leaved forest, the leaves of poplar will begin to fall off in September. After the middle of October, the soil of the site is fully exposed to the sunlight, so the soil water evaporation becomes strong, this is similar to the results of Sprenger in humid areas of England (Figure 7) [48].

### 4.3. Is Evaporation Loss f Similar to $\delta^{18}O$ Variation in Topsoil?

Studies in different climatic regions have found this shallow soil water isotope enrichment due to evaporation fractionation [32,50,62]. Compared with humid regions, the evaporation signals in arid and semi-arid environments are deeper into the soil [6,17,29,36]. Nevertheless, compared with deep soils, surface soils can respond more quickly and accurately to evaporation in different periods, and the change of evaporation loss drives the dynamics of the enrichment of soil water isotopes [13,48].

Studies in arid and semi-arid mountainous areas show that the 0 to 10 cm soil layer has high lc-excess values and evaporation loss. Sprenger conducted a similar study on the humid peatland in Scotland and found that although evaporation is very weak, the upper soil still shows an obvious fractionation signal [64]. In our research, there are significant differences in the natural environment between the top and the foothills. During the evaporation process, the low temperature and humid environment on the top of the mountain make the evaporation signal more obvious in the upper soil. The high temperature and dry environment in the piedmont area can make the evaporation signal penetrate deep into the deep soil. Furthermore, comparing the changes of 0 to 10 cm soil water isotope and evaporation loss at various sites in the study area, we found an extremely interesting rule. During the observation period, soil water $\delta^{18}O$ and evaporation loss have consistent changes. We conducted a correlation analysis between them and found that from the foothills to the top of the mountain, the correlation between soil moisture $\delta^{18}O$ and evaporation loss first strengthened and then weakened. Can we consider that the dynamics of soil water isotope enrichment driven by changes in evaporation have nothing to do with the strength of evaporation losses? There has not been a similar discussion in previous studies. Therefore, whether this conclusion is accidental or inevitable is worthy of our deep consideration (Figure 10) [57].

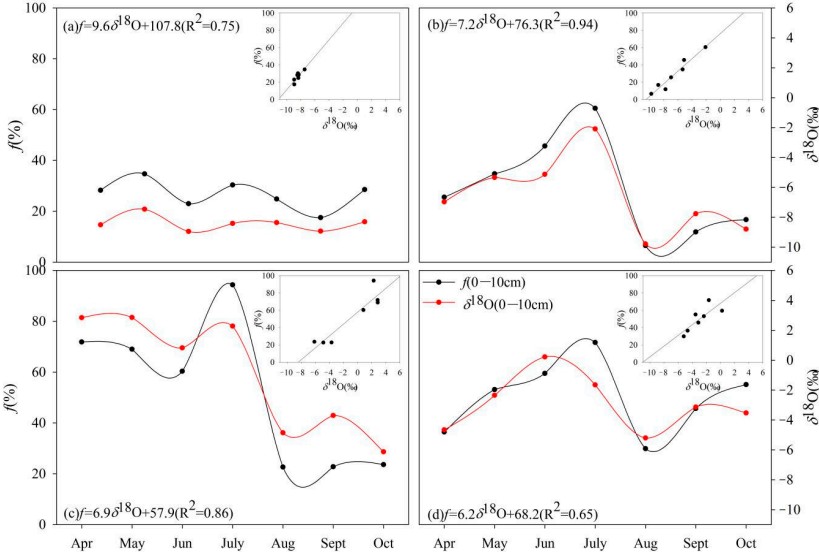

**Figure 10.** Changes of $\delta^{18}O$ and evaporation loss $f$ in 0 to 10 cm soil water at the four stations during the observation period ((**a**) Lenglong; (**b**) Ningchan; (**c**) Huajian; (**d**) Xiying).

## 5. Conclusions

Our research has made a qualitative and quantitative mutual demonstration of the stable isotope dynamics of soil water at different altitudes in arid and semi-arid alpine regions. The results prove that the change in evaporation loss drives the dynamics of the enrichment of soil water isotope. Using the lc-excess dual-isotope method to study the dynamic changes of soil water can better reflect the soil isotope enrichment caused by evaporative fractionation. In addition, the lc-excess of most soil layers at each site in the study area showed that the evaporation signal did not disappear within 90 cm (lc-excess < 0), which further proved the importance of the soil water evaporation process in arid and semi-arid environments. During the observation period, the evaporation loss of the entire region showed an upward trend before July due to the influence of temperature. After July, it was controlled by both temperature and humidity. The evaporation weakened. The vegetation type and soil salt content played important roles in the evaporation loss. In soil with low vegetation coverage, the evaporation signal of water isotopes is generally slightly higher. The results of the study may be applicable to arid and semi-arid alpine regions, and also have reference significance for changes in latitude differentiation. The study mainly emphasized the temporal and spatial heterogeneity of soil isotope evaporation signals at different altitudes, and also quantified the significant effect of evaporation loss. These results are of great value for understanding regional hydrological processes, ecological restoration services for environmentally fragile areas, and animal husbandry policy planning.

**Author Contributions:** Data curation, G.Z.; Investigation, L.Y.; Software, H.G., Z.Z., Z.S., Y.Z., Q.W., H.M., L.S., Y.L., and Y.X.; Writing—original draft preparation, L.Y.; Writing—review and editing, G.Z. All authors have read and agreed to the published version of the manuscript.

**Funding:** This research was financially supported by the National Natural Science Foundation of China (41867030, 41661005, 41761047), the National Natural Science Foundation innovation research group science foundation of China (41421061), and the Autonomous Project of State Key Laboratory of Cryosphere Sciences (SKLCS-ZZ-2017).

**Acknowledgments:** We would like to thank the colleagues in the Northwest Normal University for their help in the writing process. We are grateful to anonymous reviewers and editorial staff for their constructive and helpful suggestions.

**Conflicts of Interest:** The authors declare no conflict of interest.

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
