# Peer review of "The Soil Water Evaporation Process from Mountains Based on the Stable Isotope Composition in a Headwater Basin and Northwest China"

_water, doi:10.3390/w12102711_

Round 1
Reviewer 1 Report
This is generally a well-written paper, reporting the comprehensive isotopic dataset in soil water from arid/semi-arid inland continent. It is significant to quantitatively evaluate evaporation intensity of soil water and surface water, with concerns of different environmental and hydrological factors (e.g., T, h, P, vegetation, altitude, etc) .
I recommend the manuscript to be published, after the following moderate changes.
Line 72-74. Please check the gramma of the last sentence.
Line 107. Remove “collected”.
Line 122-123. The analytical precision should be presented more specifically. For example, “the test error” is too general. Is it standard deviation or standard error (with n=? for each sample)? Is it internal or external?
Line 136-137. “, which is of great significance … during the water cycle” is too general,so it could be removed.
Line 147. What does the “other water bodies” specifically refer to? Surface water, soil water, or leaf water?
Line 154. You need to let readers know what does “SEL” stand for.
Line 164. Please clearly document the physical meaning of k (the adjustment parameter) here.
Line 164-165. The definition of sigma is not appropriate. Sigma is usually defined as “isotopic enrichment” whereas alpha defined as “isotope fractionation factor”.
Line 180-182. Note the gramma of this sentence.
Line 186. Is this “SEL” instead of “LEL”?
Line 198. There needs space between “reflects” and “the”.
Line 201-202. As the altitude decreases, controlling factor of isotope composition of soil water include both evaporation and others, like moisture mixing, amount effect. Right? How to briefly tease apart other factors?
Table 2. Are the averaged data rainfall amount weighted, or just the average?
Line 212. It should be “secondary” evaporation.
Line 213. Note the English gramma.
Figure 4. Precision of Ic-excess is needed in the explanatory text.
Line 291. “significant differences” in water?
Line 330, 333. Report the isotopic data with precision.
Line 339, 340. How “often”, and how “sometimes”? Please put down a frequency.
Line 341. What’s the n for “a very small number”?
Line 450-451. I think it is too absolute to say “have nothing to do with”. The authors may extend some discussions on processes or environmental parameters that control the evaporation losses at the foothills vs. the top.
Author Response
Please refer to the attachment for specific modifications

Reviewer 2 Report
- Introduction is too long. Please reduce it.
- Some small text corrections.
- Figures and graphs have to be clear and readable.
- Units of each equation have to be completed (if any).
- Figure 5: why the profiles for Lenglong station are fluctuated like that? Please give short explanation the reasons behind.
- Figure 9, R2 for T and h are very low (0.24 and 0.35 respectively). Please explain it and some data are so scattered!
- Some references are very old, more than 30 years! Did you really use them? Please put only references that you really used it.

Author Response

(The authors gave the same response as above.)
